# A cell cycle-independent, conditional gene inactivation strategy for differentially tagging wild-type and mutant cells

Sonal Nagarkar-Jaiswal[1]*, Sathiya N Manivannan[2], Zhongyuan Zuo[2], Hugo J Bellen[1,2,3,4,5]*

[1]Howard Hughes Medical Institute, Baylor College of Medicine, Houston, United States; [2]Department of Molecular and Human Genetics, Baylor College of Medicine, Houston, United States; [3]Department of Neuroscience, Baylor College of Medicine, Houston, United States; [4]Program in Developmental Biology, Baylor College of Medicine, Houston, United States; [5]Jan and Dan Duncan Neurological Research Institute, Texas Children's Hospital, Houston, United States

**Abstract** Here, we describe a novel method based on intronic MiMIC insertions described in Nagarkar-Jaiswal et al. (2015) to perform conditional gene inactivation in *Drosophila*. Mosaic analysis in *Drosophila* cannot be easily performed in post-mitotic cells. We therefore, therefore, developed Flip-Flop, a *flippase*-dependent in vivo cassette-inversion method that marks wild-type cells with the endogenous EGFP-tagged protein, whereas mutant cells are marked with mCherry upon inversion. We document the ease and usefulness of this strategy in differential tagging of wild-type and mutant cells in mosaics. We use this approach to phenotypically characterize the loss of *SNF4Aγ*, encoding the γ subunit of the AMP Kinase complex. The Flip-Flop method is efficient and reliable, and permits conditional gene inactivation based on both spatial and temporal cues, in a cell cycle-, and developmental stage-independent fashion, creating a platform for systematic screens of gene function in developing and adult flies with unprecedented detail.

*For correspondence: snagarka@bcm.edu (SN-J); hbellen@bcm.edu (HJB)

## Introduction

Functional gene annotation requires, among others, the knowledge of phenotypes observed in mutants of the gene of interest, as well as the expression pattern and subcellular localization of the protein. For *Drosophila* genes whose loss causes early developmental lethality, characterization of gene function in later stages of the animal's life cycle relies on generating mosaics. Currently, Mosaic Analysis with a Repressible Cell Marker (MARCM) (*Lee and Luo, 1999*) and CRISPR/Cas9-mediated somatic mutagenesis (*Port et al., 2014*; *Xue et al., 2014*) are the two major techniques used to study mutations in mosaics. While MARCM has been very successfully used in mitotically active, developing tissues, the technique has two major limitations. First, the technique relies on mitotic recombination and can therefore not be used in non-dividing cells. Second, along with the mutation under study, other mutations in the chromosome distal to the FRT site also become homozygous in mutant cells, requiring 'rescue experiments' of the gene of interest to validate the results (*Roegiers et al., 2009*). Another strategy is based on CRISPR/Cas9-mediated somatic mutagenesis to generate mosaics (*Port et al., 2014*; *Xue et al., 2014*). The limitations of CRISPR/Cas9 are that the identity of the generated lesions for the gene of interest may vary in individual cells in the same animal or tissue. Moreover, the mutant cells are not marked and cannot be distinguished from neighboring wild-type cells.

**eLife digest** The instructions needed to build and maintain cells in an organism are encoded in their DNA. There are many different cell types, and each type only needs a small portion of the information found in the DNA to do its job. Hence, only some of the instructions, in the form of genes, need to be active or 'expressed' in any given cell type.

To understand how a gene works, it is necessary to know in which cell the gene is expressed and where in the cell the gene product – normally a protein – is located. Researchers may study a gene by deleting it, which prevents the protein from being made, or by attaching a new instruction into the gene, which generates a fluorescent tag on the protein to determine where and when it is expressed. Until now, it was not possible to selectively inactivate a gene and simultaneously mark both normal cells containing the protein and mutant cells lacking the protein.

Based on an existing tagging approach, Nagarkar-Jaiswal et al. have now developed a method in which normal and mutant cells of fruit flies are marked differently. A gene of interest is tagged with a fluorescent marker called green fluorescent protein (or GFP). The same gene is then inactivated in some of the cells, which are tagged with a red marker called mCherry. Nagarkar-Jaiswal et al. compared normal and mutant cells, and were able to determine how long it takes before the mutant cells become abnormal.

With this new method, the role of numerous genes in any tissue of adult flies can be reassessed. This will allow to investigate what happens when a protein is removed in specific cells in adult flies. A future goal will be to apply this method to other animals that are more closely related to humans, such as mice, to gain a clearer picture of the role of genes in different cell types and how faulty genes may cause disease.

Here, we describe a new flippase (FLP)-dependent method 'Flip-Flop' that offers several advantages over the current techniques for generating mosaics. (1) The method does not rely on cell division and can, therefore, be broadly used for conditional gene inactivation in post-mitotic cells such as neurons. (2) It allows endogenous tagging of proteins with EGFP, which permits multiple applications, and (3) it simultaneously marks mutant cells with mCherry.

## Results and discussion

### The Flip-Flop construct and strategy

We engineered the 'Flip-Flop' cassette for conditional gene inactivation. This cassette contains two modules that are placed in opposite orientation: a protein-trap (PT) module and a gene-trap (GT) module (*Figure 1A*). The PT module carries a splice acceptor (SA), followed by an in-frame EGFP coding sequence, and a splice donor (SD) (*Venken et al., 2011*). The GT module similarly contains a SA, but is followed by the T2A peptide sequence, an mCherry coding sequence, stop codons in all three reading frames, and an SV40 polyA signal. The PT and GT modules are placed in opposite orientations and are flanked by inverted pairs of canonical FRT and FRT14 sites, forming a flip-excision switch (FLEx) (*Schnütgen et al., 2003*; *Xue et al., 2014*). The entire cassette is nested between two inverted *attB* sites that facilitate Recombination-Mediated Cassette Exchange (RMCE) between the Flip-Flop cassette and a target Minos-Mediated Integration Cassette (MiMIC) that resides in a coding intron of the gene of interest. When integrated in the MiMIC in the PT orientation, Flip-Flop should result in expression of the endogenous protein with an internal EGFP tag. The internal EGFP tagging does not or subtly disrupt the protein's function in 77% of the cases (*Nagarkar-Jaiswal et al., 2015*). The cassette can then be converted from a PT to a GT, in vivo, by inverting the cassette's orientation through the expression of *FLP* that acts on the FLEx switch (*Figure 1B*). Following the switch from the PT to the GT orientation, transcription is precociously terminated by the polyA sequence. When this truncated transcript is translated, the T2A site induces a translational skip (*Tang et al., 2009*), truncating the native protein and re-initiating translation at the mCherry sequence (*Figure 1B*). Hence, inversion of the Flip-Flop cassette results in the generation of a truncated protein, which is typically non-functional, and simultaneously marks the cells that are actively

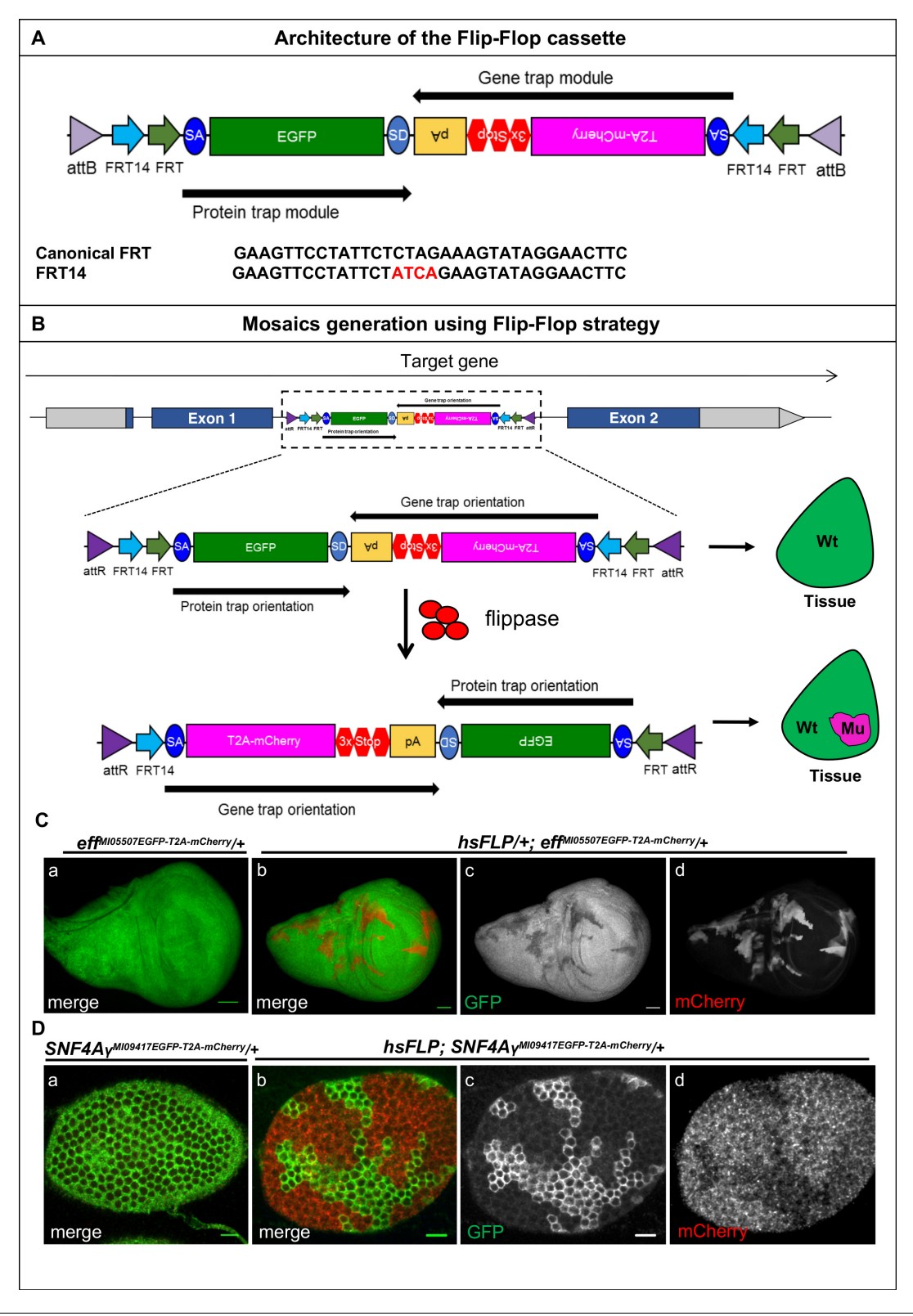

**Figure 1.** Mosaic generation using the Flip-Flop cassette. (**A**) The architecture of the Flip-Flop cassette. The cassette consists of two independent modules (PT and GT), that are oriented in opposite orientations. The PT module contains a splice acceptor (SA), followed by an EGFP tag and a splice donor (SD). The GT module contains a SA sequence, followed by the T2A peptide coding sequence (which will induce a translational skip), the mCherry coding region, stop codons in all three coding frames, and an SV40 polyA transcriptional termination signal. Given the opposite orientation of both

*Figure 1 continued on next page*

*Figure 1 continued*

modules, only one of the SA sequence will be active with respect to the recipient gene. The two modules are nested within a pair of *FRT* and *FRT14* inverted repeats, forming a flippase-responsive FLEx switch. Finally, the entire cassette is flanked by two inverted *attB* sequences that permit *phiC31*-mediated RMCE between the Flip-Flop cassette and pre-existing MiMIC elements. A comparison of the *FRT* and *FRT14* sequence is shown below. The *FRT14* sequence varies from the canonical *FRT* sequence at the residues highlighted in red. (**B**) Schematic showing the inversion of a PT-oriented Flip-Flop cassette, inserted into the coding intron of a hypothetical gene. Upon *FLP*-expression, the FLEx switch undergoes two recombination events: (1) recombination between the two *FRT* sites or between the two *FRT14* sites leads to cassette inversion that is followed by (2) excision of either the pair of *FRT* sites or the pair of *FRT14* sites, both of which have obtained the same orientation due to the flip. Since the remaining unpaired *FRT* and *FRT14* sites are not able to recombine, the cassette will be locked in the GT orientation. Thus, the initial PT orientation allows the gene to be tracked by EGFP-tagged protein expression in tissues. *FLP* activity inverts the Flip-Flop cassette in random cells, generating a mosaic tissue consisting of cells that did not undergo the flip and are still expressing the EGFP-tagged protein and cells that inverted the Flip-Flop cassette into the GT orientation, which are marked by mCherry expression. (**C**) (a) Wing imaginal disc from a *eff*$^{MI05507EGFP-T2A-mCherry}$/+ third instar larva, expressing Eff-EGFP-Eff (b) Wing imaginal disc from a *hsFLP; eff*$^{MI05507EGFP-T2A-mCherry}$/+ third instar larva showing *hsFLP*-induced clones that loose the GFP signal (c) and express mCherry (d) Scale bar = 50 µm. (**D**) (a) Stage 6 egg chamber obtained from *SNF4Aγ*$^{MI09417EGFP-T2A-mCherry}$/+ females reveals EGFP-SNF4Aγ-EGFP expression in all follicle cells. (b) Stage 6 egg chamber obtained from *hsFLP; SNF4Aγ*$^{MI09417EGFP-T2A-mCherry}$/+ females shows a mosaic tissue with groups of cells that have retained EGFP expression, and others that have inverted the Flip-Flop cassette and can be recognized by loss of EGFP expression (c) and gain of mCherry expression (d), inversion of Flip-Flop cassette is marked with loss of EGFP and expression of mCherry. Scale bar = 10 µm.

The following source data and figure supplements are available for figure 1:

**Source data 1.** List of constructs generated in this study.

**Source data 2.** List of fly strains used in the study.

**Figure supplement 1.** Flip-Flop PT insertions in *eff* and *SNF4Aγ* generate functional EGFP-tagged proteins.

**Figure supplement 2.** Crossing schemes for FLP-mediated conditional gene inactivation.

**Figure supplement 3.** Inversion of the Flip-Flop PT insertion in *Cdep* using *ey-FLP*.

**Figure supplement 4.** Inversion of GT-oriented Flip-Flop insertions.

**Figure supplement 5.** The efficiency of Flip-Flop insertions.

transcribing the gene with mCherry. While mCherry's expression pattern recapitulates the spatio-temporal expression pattern of the recipient gene, it does not reproduce the endogenous subcellular localization of the protein (*Figure 1B*). In summary, expression of *FLP* will induce a Flip-Flop and produce mCherry-labeled mutant cells in which the gene is inactivated, whereas the surrounding cells are wild-type and express the EGFP-tagged protein.

## Flip-Flop-mediated mosaics in mitotically active cells

We created PT insertions for nine genes using available MiMIC insertions: *effete* (*eff*), *Nedd8*, *SNF4/AMP-activated protein kinase gamma subunit* (*SNF4Aγ*) *Chondrocyte-derived ezrin-like domain containing protein* (*Cdep*), *Tripartite motif containing 9* (*Trim9*), *Salt-Induced kinase3* (*Sik3*), *Ankyrin2* (*Ank2*), *Circadian trip* (*Ctrip*) and *Ecdysone-induced protein 63E* (*Eip63E*). Seven of the nine generated PT insertions complement the lethality associated with loss of the corresponding gene, suggesting that these internally tagged proteins are biologically functional, consistent with previous observations (*Nagarkar-Jaiswal et al., 2015*; *Venken et al., 2011*). The PT insertion in *eff* (*Figure 1—figure supplement 1A*) introduces the expression of internally tagged Eff (Eff-EGFP-Eff) in the wing discs of *eff*$^{MI05507-EGFP-T2A-mCherry}$ larvae (*Figure 1C* a). Heat-shock-induced expression of *FLP* (*hsFLP*) (*Figure 1—figure supplement 2A*) causes loss of the EGFP signal and induces mCherry expression in random subsets of cells (*Figure 1C* b-d). Similarly, the PT insertion in *SNF4Aγ* (*Figure 1—figure supplement 1B* a) leads to SNF4Aγ-EGFP-SNF4Aγ expression in adult egg chambers (*Figure 1D* a). The SNF4Aγ Flip-Flop cassette can be flipped to the GT orientation efficiently, generating large clones of cells expressing mCherry upon heat-shock (*Figure 1D* b-d). Finally, we tagged *Cdep* (*Figure 1—figure supplement 3A* a-b), observed EGFP expression in the eye-antennal discs,

and efficiently induced inversions using *ey-FLP* (*Figure 1—figure supplement 3B* and *Figure 1—figure supplement 2B*). Hence, the PT orientation insertions permit gene expression analysis and can effectively be inverted to create mCherry marked mutant cells.

A comparison of the PTs generated using a previous, shorter, RMCE construct (GFSTF; *Venken et al., 2011*) and the PT of the Flip-Flop cassette did not show any obvious difference in expression pattern or genetic properties based on complementation tests. In summary, if the protein traps are functional with the GFSTF cassette they are also functional with the EGFP-tagged proteins derived from the Flip-Flop.

In addition to the above-described PT lines, we generated GT-lines expressing mCherry for five genes: *eff*, *Cdep*, *Trim9*, *Ank2*, and *Ctrip*. Complementation tests with independently derived loss-of-function alleles (or deficiencies) indicate that these GT-alleles are indeed loss-of-function alleles. Note that none of these GT insertions induce an obvious dominant phenotype. We observed efficient conversion of an initial GT insertion to the PT orientation for *eff*$^{MI05507\ T2A-mCherry-EGFP}$, using either the *ey-FLP* or *hsFLP* drivers in larval brains. As shown, the use of *ey-FLP* introduces Eff-EGFP-Eff expression in the eye discs (area within white dotted lines), as well as the area of the brain where *eyeless* is expressed (demarcated by blue dotted lines) (*Figure 1—figure supplement 4A*). In contrast to the *ey-FLP*, smaller clones of randomly positioned wild-type and mutant cells can be generated by stochastic FLP expression when applying heat shocks. Indeed, heat-shock-induced *hsFLP* expression brought about small patches of EGFP-positive cells in *eff*$^{MI05507\ T2A-mCherry-EGFP}$ larval brains (*Figure 1—figure supplement 4B*). This type of experiment creates wild-type cells in an otherwise mutant animal and should help to assess whether a mutant phenotype can be attenuated or reverted, or not. In addition, these experiments may help identify in which tissue an essential gene is required.

Finally, for *Nedd8* and *Sik3* we generated PT lines, but these insertions failed to complement corresponding deficiencies, showing that the internal EGFP tag disrupts protein function. Nevertheless, we were able to induce inversions for both genes using *ey-FLP*, confirming our observation that flipping the Flip-Flop cassette is efficient (*Figure 1—figure supplement 4C* and *Figure 1—figure supplement 5D and E*). In summary, both orientations of the Flip-Flop cassette could effectively be flipped.

To examine and compare the efficiency of the Flip-Flop inversions, we tested six genes using the same FLP-source, under identical conditions. For *SNF4Aγ*, *Trim9* and *eff*, more than 95% of the cells undergo cassette inversion. For *Nedd8*, *cdep*, and *Sik3*, more than 70% cells displayed Flip-Flop inversions (*Figure 1—figure supplement 5*). Hence, inducing Flip-Flop inversions in eye-antennal discs is highly efficient for all genes tested.

## Conditional gene inactivation using Flip-Flop in post-mitotic cells

Next, we tested if Flip-Flop can be used for gene inactivation in post-mitotic cells by using *Trim9* as an example. Previously, *Song et al. (2011)* created an imprecise excision of a P-element (P{GawB} Trim9[NP4638]) inserted in the 5′-regulatory region of *Trim9* and reported that animals lacking the first few amino acids of Trim9 are viable and exhibit a droopy wing phenotype (*Song et al., 2011*). We generated both GT- (*Trim9*$^{MI12525P\ T2A-mCherry-EGFP}$) and PT-oriented (*Trim9*$^{MI12525EGFP-\ T2A-mCherry}$) Flip-Flop insertions in *Trim9* (*Figure 2—figure supplement 1A* a). The GT insertion is homozygous lethal and does not complement a deficiency (*Trim9*$^{Df}$), while the PT insertion is viable in trans over *Trim9*$^{Df}$ (*Figure 2—figure supplement 1A* b) suggesting that the PT encodes a functional protein. When the PT insertion in *Trim9*$^{MI12525EGFP-\ T2A-mCherry}$/*Trim9*$^{Df}$ animals was inverted in all cells using a ubiquitously expressed (*Actin-GAL*) FLP, the animals died as second instar larvae. This, together with the lethality observed in the GT insertion, suggests that a strong loss of *Trim9* leads to early developmental lethality.

We then examined the consequences of neuronal loss of *Trim9*, by inverting the *Trim9*$^{MI12525EGFP\ T2A–mCherry}$ PT insertion in larval and adult brains using neuronally expressed FLP (*nSyb-GAL4>UAS-FLP*) (*Figure 1—figure supplement 2C* and *Figure 2—figure supplement 1B*). Prior to the flip, in the PT orientation, *Trim9* is expressed in numerous neurons in the larval brain (*Figure 2A* a-d) and the protein is localised to nuclei and cytoplasm (*Figure 2A* e-h). These cells do not express mCherry. Following inversion to the GT orientation, we detected mCherry expressing cells in the larval brain (*Figure 2A* a′-d′). Upon close examination, we observed three different populations of neurons. First, cells that have undergone cassette inversion and are marked with mCherry (*Figure 2A* e′-h′,

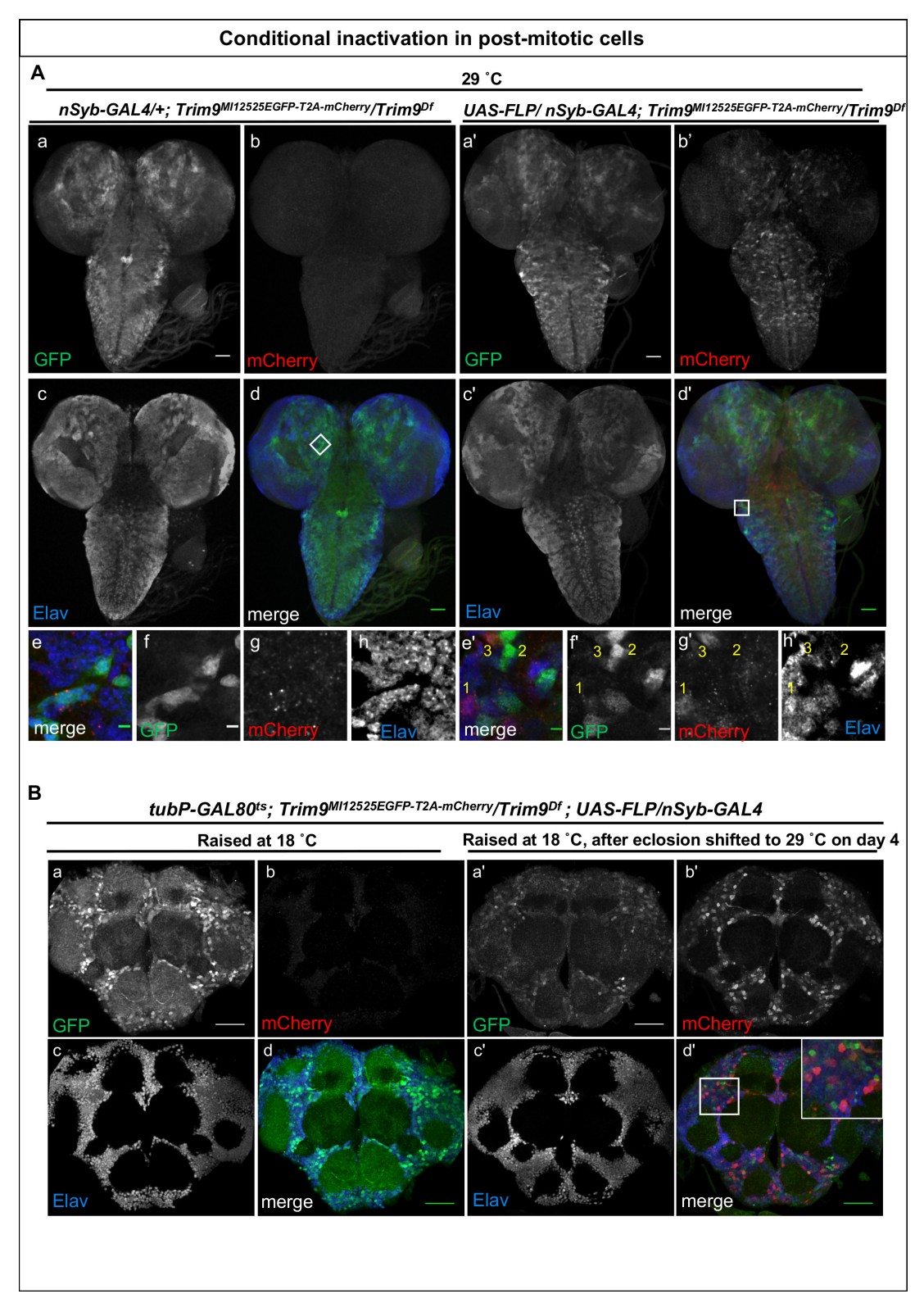

**Figure 2.** Inactivation of *Trim9* in post-mitotic cells using *nSyb-GAL4/UAS-FLP*. (**A**) A third instar larval brain of *nSyb-GAL4/+; Trim9^{MI12525EGFP-T2A-mCherry}/Trim9^{Df}*, raised at 29°C stained for EGFP (green, a, d and f), mCherry (red, b, d and g) and Elav (blue, c, d and h). In the absence of UAS-*FLP*, only GFP signal is detected, reflecting the Trim9-EGFP-Trim9 expression pattern. In contrast, mCherry is not expressed. (b, d and g). Magnification of the region indicated by a white square in panel d (e–h). Third instar larval brains of *UAS-FLP/nSyb-GAL4; Trim9^{MI12525EGFP-T2A-mCherry}/Trim9^{Df}*,

*Figure 2 continued on next page*

Nagarkar-Jaiswal *et al*. eLife 2017;6:e26420. DOI: 10.7554/eLife.26420

Figure 2 continued

expressing *FLP* under the control of *nSyb-GAL4* show cassette inversion that leads to the loss of EGFP and gain of mCherry expression in neurons (a'–h'). Magnification of the region indicated by a white square in panel (d') shows the existence of three populations of neurons (e'–h'): neurons that express mCherry and have completely lost GFP expression (1), neurons that have not undergone cassette inversion and thus still express GFP but lack mCherry expression (2) and neurons that express both tags, suggesting they have recently undergone cassette flip (as indicated by mCherry expression), and still contain some remaining Trim9-EGFP-Trim9 protein that perdures following the cassette flip, scale bar = 50 μm (a, b, c, d, a', b', c' and d'), scale bar = 5 μm (e, f, g, h, e', f', g' and h'). (B) Adult brain of *tubP-GAL80^{ts}; Trim9^{MI12525EGFP-T2A-mCherry}/Trim9^{Df}; UAS-FLP/nSyb-GAL4* animals stained for EGFP (green), mCherry (red) and Elav (blue). The temperature-sensitive *GAL80* (*GAL80^{ts}*), expressed under a tubulin promoter (*tubP*) was used to prevent *nSyb-GAL4* from driving expression of *UAS-FLP* during development. (a–d) Adult brain of *tubP-GAL80^{ts}; Trim9^{MI12525EGFP-T2A-mCherry}/Trim9^{Df}; UAS-FLP/nSyb-GAL4* control animals, raised at 18°C display Trim9-EGFP-Trim9 (green, a and d) and do not show any mCherry expression (red, b and d). (a'–d') Adult brains of *tubP-GAL80^{ts}; Trim9^{MI12525EGFP-T2A-mCherry}/Trim9^{Df}; UAS-FLP/nSyb-GAL4* animals, raised at 18°C until 4 days after eclosion, and then shifted to 29°C to induce *FLP*-expression, show mCherry-expressing neuronal cells in which the cassette inversion has taken place (red, b' and d'), scale bar = 50 μm.

The following figure supplement is available for figure 2:

**Figure supplement 1.** Inactivation of *Trim9* in post-mitotic cells through induction of *FLP* using *nSyb-GAL4/UAS-FLP*.

indicated by number (1) Second, cells that are EGFP-positive (Trim9-EGFP-Trim9) and did not flip (*Figure 2A* e'-h', indicated by number (2) Third, a population of cells that expresses both Trim9-EGFP-Trim9 as well as mCherry (*Figure 2A* e'-h', indicated by number (3) Note that Trim9-EGFP-Trim9 levels are lower in these cells. We reason that these cells have undergone cassette inversion and are hence expressing mCherry, but that the Trim9-EGFP-Trim9 that was produced prior to the flip has not been fully degraded. Thus, the population of cells that express only mCherry must have lost most of the tagged Trim9 protein, whereas Trim9-EGFP-Trim9 perdures in a subset of cells that are yellow. Importantly, this allows analysis of protein perdurance. In addition, it permits researchers to only select and analyse those cells that are truly mutant and have lost the majority of the wild-type protein. Neuronal *Trim9* inactivation does not affect the eclosion of the animal but leads to the sterility of short-lived animals with droopy wings, corroborating previous observations associated with the loss of *Trim9* (*Song et al., 2011*).

Next, we explored the use of the Flip-Flop cassette in post-mitotic, adult neurons. We expressed GAL80^{ts} (a suppressor of GAL4 at low temperature) to prevent *nSyb-GAL4* from inducing flips during development. *tubP-GAL80^{ts}; Trim9^{MI12525EGFP- T2A-mCherry}/Trim9^{Df}; UAS-FLP/nSyb-GAL4* animals were raised at 18°C until the fourth day of adulthood and were subsequently shifted to 29°C. Animals that were not shifted to 29°C did not express mCherry in adult brains (*Figure 2B* a-d). In contrast, those shifted to a higher temperature on the fourth day of adulthood showed loss of Trim9-EGFP-Trim9 expression and gain of mCherry in a large subset of neurons (*Figure 2B* a'-d'). Interestingly, animals shifted to 29°C do not show droopy wing phenotype, but have a reduced lifespan. This suggests that the droopy wing phenotype observed by *Song et al. (2011)* is due to the loss of neuronal *Trim9* during development, and not because of loss of the gene in adults.

## Loss of *SNF4Aγ* in adults flies leads to severe neurodegeneration

To further explore the use of Flip-Flop in adult flies, we selected a subunit of the AMP-activated protein kinase (AMPK) complex. The AMPK complex is involved in sensing stresses such as a drop in the ratio of ATP/AMP, hypoxia, ischemia, and heat-shock (*Hardie et al., 2003*). The ATP/AMP ratio is sensed through an allosteric mechanism by the non-catalytic γ subunit of the AMPK complex, which in turn promotes the phosphorylation of the catalytic α subunit enhancing α subunit's kinase activity (*Sanders et al., 2007*; *Scott et al., 2004*). Phosphorylation of AMPK's targets by the α-subunit activates a signalling cascade that ultimately regulates fatty acid oxidation, autophagy, and mitochondrial biogenesis and boosts the ATP/AMP ratio within the cell (*Hardie et al., 2012*). Sensing the ATP/AMP ratio by the γ subunit is thus critical for the proper function of the AMPK complex, and together with the other AMPK subunits, the γ subunit is crucial for proper energy homeostasis (*Hardie et al., 2003*).

In *Drosophila*, the γ subunit is encoded by *SNF4Aγ*, with 16 transcriptional isoforms. Of these, 15 should be tagged using MI09417 (*Figure 1—figure supplement 1B* a). We converted this MiMIC

insertion into a T2A-GAL4 (*Diao et al., 2015*) allele (*SNF4Aγ*^MI09417T2A>GAL4^), and a Flip-Flop insertion in the PT orientation (*SNF4Aγ*^MI09417EGFP- T2A-mCherry^) (*Figure 3A* and *Figure 1—figure supplement 1B* a). *SNF4Aγ*^MI09417T2A>GAL4^ truncates the SNF4Aγ protein and simultaneously expresses GAL4 in the spatiotemporal pattern of *SNF4Aγ*. The *SNF4Aγ*^MI09417 T2A>GAL4^ allele fails to complement the *SNF4Aγ*^Df^ and acts as a loss-of-function allele. On the other hand, *SNF4Aγ*^MI09417EGFP-T2A-mCherry^/*SNF4Aγ*^MI09417^ and *SNF4Aγ*^MI09417EGFP-T2A-mCherry^/*SNF4Aγ*^Df^ animals are viable, indicating that the Flip-Flop insertion produces a functional SNF4Aγ-EGFP-SNF4Aγ protein (*Figure 1—figure supplement 1B* b).

In *SNF4Aγ*^MI09147 T2A>GAL4^ /*SNF4Aγ*^MI09147EGFP- T2A-mCherry^ animals, SNF4γ-EGFP-SNF4Aγ is broadly expressed in larvae (data not shown). In the presence of *UAS-FLP*, the GAL4 from *SNF4Aγ*^MI09417 T2A>GAL4^ will drive FLP expression in cells expressing SNF4Aγ, which will convert the PT insertion into a GT insertion. The GT insertion, while expressing mCherry, leads to the loss of functional protein produced by the *SNF4Aγ*^MI09147EGFP-T2A-mCherry^ allele. Hence, *UAS-FLP;; SNF4Aγ*^MI09417EGFP-T2A-mCherry^/*SNF4Aγ*^MI094177 T2A>GAL4^ animals lack expression of both alleles of *SNF4Aγ* (*Figure 1—figure supplement 2D*). These animals die as poorly developed pupae and are slim and elongated in the larval stage (*Figure 3B* b-c). These larvae show a broad expression of mCherry that resembles the EGFP expression pattern from *SNF4Aγ*^MI09417EGFP- T2A-mCherry^ allele (*Figure 3B* b).

Next, we induced Flip-Flop inversion in eyes. The *Drosophila* eye is a highly organised structure that is composed of about 800 small functional units called ommatidia. Every ommatidium contains eight photoreceptor cells (R1-R7) each with a rhabdomere, the light-sensing organelle. Within each ommatidium, rhabdomeres are arranged in a highly stereotypic fashion (*Figure 3E* c control panel, indicated by numbers). In *ey-FLP;;SNF4Aγ*^MI09417EGFP-T2A-mCherry^/*SNF4Aγ*^Df^ animals, almost all photoreceptors inverted from the PT- to GT- orientation, as revealed by mCherry expression (*Figure 3E* b'). Inverting the cassette in *SNF4Aγ* to a GT orientation reduces the amplitude of the electroretinogram (ERG) in 1-day-old flies. These animals did not show a stronger reduction in ERG amplitude when measured at day 30 and hence do not show further degeneration (*Figure 3C and D*). We also observed a reduction in the number of rhabdomeres as well as defects in photoreceptor organisation (*Figure 3E–c'*). The loss of SNF4Aγ disrupts energy homeostasis, possibly prevents the photoreceptors from meeting their high-energy demand and leading to a reduction in their activity. Similar defects have been described in *alicorn* mutant eye clones (SNF4Aα) (*Spasić et al., 2008*).

It has been previously shown that *SNF4Aγ* is required for autophagy (*Lippai et al., 2008*). We, therefore, analysed the role of *SNF4Aγ* in autophagy in mutant cells generated by Flip-Flop inversion. To this end, we induced Flip-Flop-mediated loss of *SNF4Aγ* in the adult gut and starved adults for 24 hr by growing them on agar lacking amino acids (aa) (*Figure 4A*). To analyse the induction of autophagy, we stained adult gut from these animals for p62, a marker that is degraded by autophagy. If *SNF4Aγ* is required for autophagy, loss of SNF4Aγ should lead to p62 accumulation in mutant cells, as they should not induce autophagy. However, we observed a loss of p62 in mutant cells (mCherry positive) and robust staining of p62 in control cells (EGFP positive, indicated by red arrows) (*Figure 4B* and *Figure 4—figure supplement 1*). Interestingly, this parallels the observation by *Sahani et al. (2014)* in vertebrate cells. These authors observed that following prolonged amino acid starvation, the autophagy marker p62 is restored because of an increase in transcription and translation mediated by the availability of autophagy derived aa. Hence, we propose that loss of *SNF4Aγ* blocks autophagy, deprives cells of aa during starvation, leading to a loss of p62. Similar data have been documented in *Drosophila* larvae and mammalian cells (*B'chir et al., 2014*; *Colosetti et al., 2009*; *Duran et al., 2011*; *Erdi et al., 2012*).

Previous studies have described that developmental loss of *SNF4Aγ* leads to neurodegeneration (*Cook et al., 2012*; *Tschäpe et al., 2002*). To further characterise the role of *SNF4Aγ* in *Drosophila* neurons, we used a post-mitotic GAL4 driver *C155-GAL4*, to drive *UAS-FLP* and obtain GT orientated Flip Flops in all neurons. Such pan-neuronal flip of *SNF4Aγ*^MI09417EGFP-T2A-mCherry^ into a nonfunctional GT leads to severe locomotor defects in adult flies (data not shown). We also examined the role of *SNF4Aγ* in adults using *hsFLP*-mediated cassette inversion. Flies of the genotype *hsFLP; SNF4Aγ*^MI09417EGFP- T2A-mCherry^ / *SNF4Aγ*^MI09417^ were raised at 18°C and were given a 2-hr heat-shock at 37°C, 3 days after eclosion. Two days after the heat-shock, H and E staining of adult brains revealed severe neurodegeneration. The extent of neuronal loss did not worsen by day 10 (*Figure 4C* b and d). In addition, these flies typically die 3 to 4 weeks later. Following heat-shock,

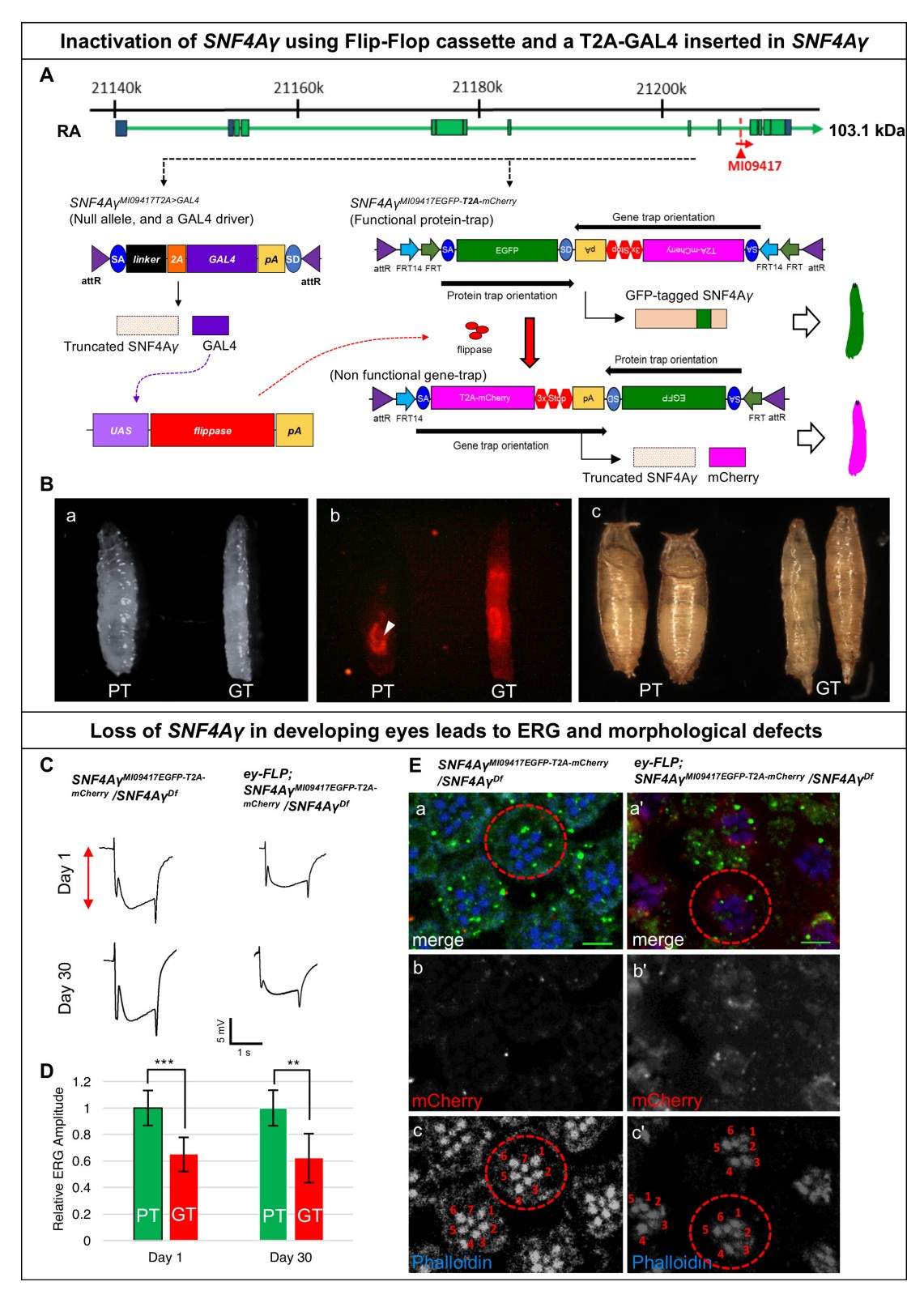

**Figure 3.** Developmental and neuronal functions of *SNF4Aγ* revealed through broad or tissue-specific inactivation. (**A**) Gene structure of *SNF4Aγ*, displaying one of the transcriptional isoforms and the precise location of MiMIC insertion MI09417 (red triangle) as well as the orientation of the MiMIC cassette (red arrow). The structure of the *T2A-GAL4* insertion (*SNF4γ^MI09417T2>GALl4*) and the PT oriented Flip-Flop insertion (*SNF4Aγ^MI09417EGFP-T2A-mCherry*) are shown below. (**B**) (a) Control third instar larva (*SNF4Aγ^MI09417EGFP-T2A-mCherry*/*SNF4Aγ^MI09417 T2A>-GALl4*; PT) and third instar larva in which

*Figure 3 continued on next page*

Figure 3 continued

SNF4Aγ is inactivated through cassette flip, driven by the *T2A-GAL4* insertion in *SNF4Aγ* on the other chromosome (*UAS-FLP;; SNF4Aγ$^{MI094177EGFP-T2A-mCherry}$/SNF4Aγ$^{MI09417T2A>GAL4}$;* GT) (b) Expression of mCherry in larvae shown in (a). Arrowhead indicates auto-fluorescence of gut tissue in control animals (*SNF4Aγ$^{MI04147EGFP-T2A-mCherry}$/SNF4Aγ$^{MI09417T2A-GAL4}$;* PT), which differs from the mCherry signal in animals that underwent cassette flip (*UAS-FLP;;SNF4Aγ$^{MI09147EGFP-T2A-mCherry}$/SNF4Aγ$^{MI09147\ T2A>GAL4}$,* GT). (c) Pupae of control (PT) and experiment genotypes (GT). Loss of *SNF4Aγ* leads to premature pupariation and death. (C) *ey-FLP*-mediated cassette inversion in developing eyes leads to electroretinogram (ERG) defects. (Left) ERG trace obtained from control animals (*SNF4Aγ$^{MI09417EGFP-T2A-mCherry}$/SNF4Aγ$^{Df}$*) on day 1 and day 30. (Right) ERG trace obtained from experimental animals (*ey-FLP; SNF4Aγ$^{MI09417EGFP-T2A-mCherry}$/SNF4Aγ$^{Df}$*) in which *ey-FLP*-mediated cassette inversion induced loss of *SNF4Aγ* in >95% of the developing eye (see also **Figure 1—figure supplement 5B**). (D) Histogram showing relative ERG amplitude (indicated by the red double-headed arrow in (**C**)), measured on days 1 and 30 of control animals (PT/green) and experimental animals with *ey-FLP*-mediated *SNF4Aγ* inactivation (GT/red). (At least eight animals of each genotype were analysed. *** indicates a p-value<0.001 and ** indicates a p-value<0.01 obtained by performing a student T-test). (E) Image displaying adult eyes of control (*SNF4Aγ$^{MI09417EGFP-T2A-mCherry}$/SNF4Aγ$^{Df}$,* a–c) and experimental (*ey-FLP; SNF4Aγ$^{MI09417EGFP-T2A-mCherry}$/SNF4Aγ$^{Df}$,* a'–c') animals stained with phalloidin (blue) marking the seven visible photoreceptors arranged within each ommatidium (red dotted circle), SNF4Aγ-EGFP-SNF4Aγ expression (green) and mCherry expression (red). (b, b', c and c') mCherry channel (b, b') and phalloidin staining (c, c') of the image shown in (a and a'). (a') *eyFLP*-driven loss of *SNF4Aγ* induces mCherry expression (b') and leads to defects in photoreceptor arrangement and loss of individual rhabdomeres in ommatidia (red dotted circle) (c' indicated by numbers), whereas control ommatidia consistently contain seven visible rhabdomeres (c), scale bar = 5 μm.

animals that do not carry *hsFLP* do not show obvious lesions in the brain (**Figure 4C** a and c). We conclude that loss of *SNF4Aγ* leads to neuronal demise in adult flies.

In conclusion, the Flip-Flop strategy allows conditional gene inactivation and generation of mosaics for a side-by-side comparison of mutant and wild-type cells in the same tissue, while simultaneously marking wild-type and mutant cells with different fluorescent markers. By using different sources of *FLP*, one can control the spatial and temporal pattern of mosaic generation and gene inactivation. Using *hsFLP*, we show that Flip-Flop can be used to severely inhibit gene function during development, as well as in adults in both mitotically active and post-mitotic cells. Flip-Flop can therefore reliably be used to explore gene functions both during development and in adult flies. Recently, a similar Flippase-based conditional gene inactivation method 'FlpStop' was published (**Fisher et al., 2017**). Both the FlpStop and Flip-Flop strategies employ a similar Flex switch to induce local cassette inversion. However, there are several important differences between the two technologies. (1) FlpStop does not tag the endogenous protein with a fluorescent tag, while Flip-Flop produces an internally EGFP-tagged protein. While this may seem beneficial for genes where an internal tag is deleterious to the protein, we argue that for the majority of the genes the internal EGFP tag is not detrimental (this study; **Nagarkar-Jaiswal et al., 2015**). In contrast, endogenous tagging is very advantageous as it permits to track the expression and subcellular localization of a protein, allows anti-GFP antibody-mediated immunoprecipitations followed by mass spectroscopy (**David-Morrison et al., 2016**; **Yoon et al., 2017**), enables ChIP sequencing (**Nègre et al., 2011**), and permits mRNA or protein removal with iGFPi (**Neumüller et al., 2012**) or deGradFP (**Caussinus et al., 2011**; **Nagarkar-Jaiswal et al., 2015**; **Urban et al., 2014**). Finally, the EGFP tag allows us for the first time to track protein perdurance in mutant cells. (2) The second major difference between FlpStop and Flip-Flop relates to how mutant cells are marked. In FlpStop, tdTomato is expressed under the control of the UAS/GAL4 system. Hence, cells that have undergone inversion will express tdTomato, irrespective of the expression pattern of the gene. In contrast, Flip-Flop will label cells with mCherry only if they express the gene of interest. In summary, the ability to track the loss of the protein of interest in mutant cells concomitant with loss of the EGFP signal while simultaneously marking the mutant cells with mCherry from the same regulatory elements distinguishes Flip-Flop from all other mosaic analyses methods.

We would like to note that Flip-Flop insertions can be generated based on existing MiMIC insertions (**Nagarkar-Jaiswal et al., 2015**) as well as MiMIC-like elements introduced by CRISPR/Cas9 (**Zhang et al., 2014**) and can be extended to *Drosophila* cell culture system using RMCE (**Manivannan et al., 2015**). We, therefore, believe that the use of Flip-Flop will permit functional annotation of numerous genes in unprecedented detail.

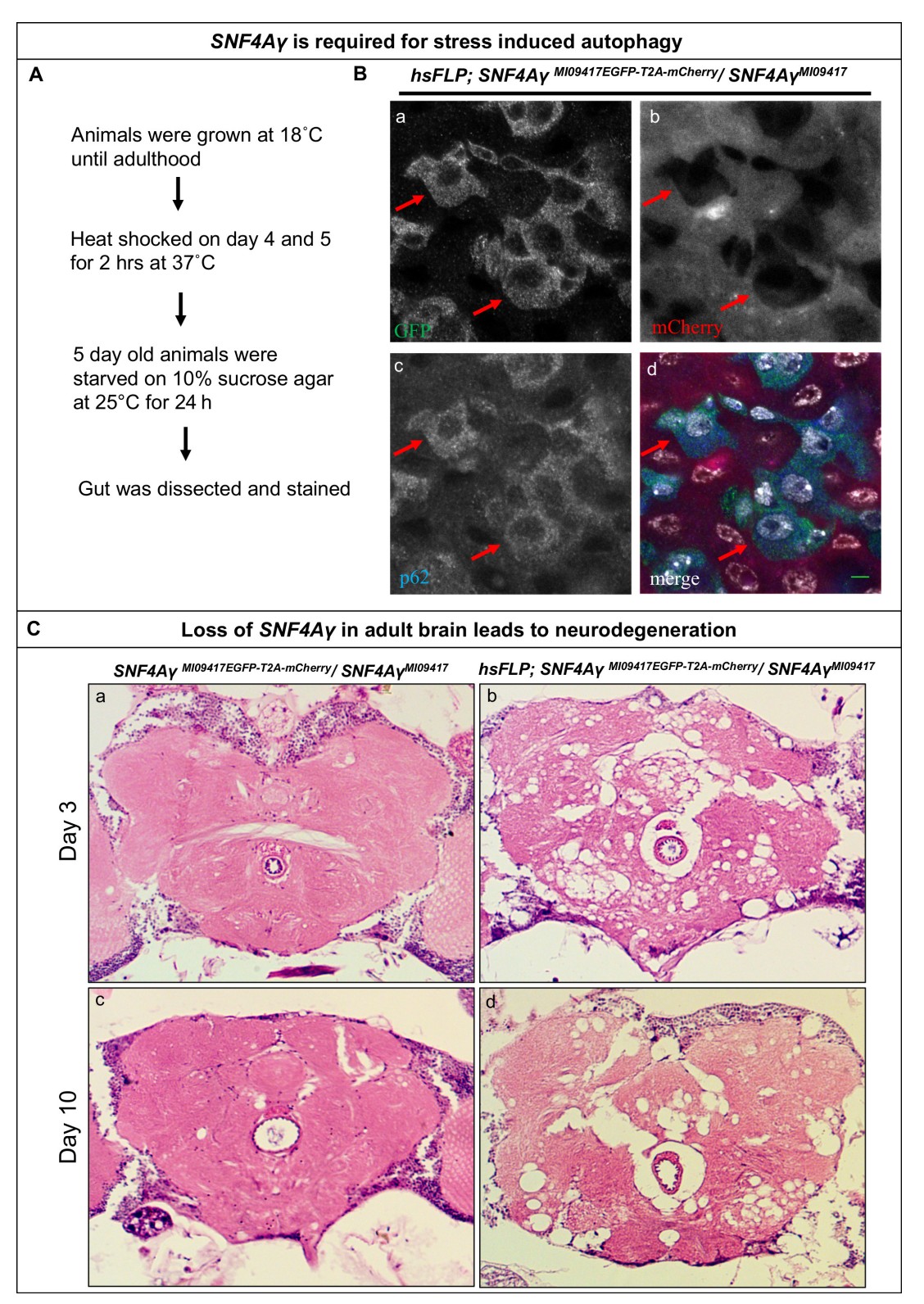

**Figure 4.** Loss of *SNF4Aγ* in adults flies leads to autophagic and neurodegenerative phenotypes. (**A**) Flow chart describing the experimental outline to induce autophagy in the adult gut (**B**) Image displaying the adult midgut from amino acid-starved *hsFLP; SNF4Aγ^MI09417EGFP-T2A-mCherry^/* SNF4Aγ^MI09417^ animals, stained for GFP (a, d), mCherry (b, d), p62 (c, d) and DAPI (white) (d). Red arrows indicate control cells expressing SNF4Aγ-EGFP-SNF4Aγ, revealing higher levels of p62. In contrast, mutant cells are mCherry positive and have reduced p62 levels, indicating a defect in autophagy induction,
*Figure 4 continued on next page*

*Figure 4 continued*

scale bar = 5 µm (C) Haemotoxylin and Eosin staining on brain sections of *SNF4Aγ*$^{MI09417EGFP\text{-}T2A\text{-}mCherry}$/*SNF4Aγ*$^{9417}$ (control) and *hsFLP*; *SNF4Aγ*$^{MI09417EGFP\text{-}T2A\text{-}mCherry}$/ *SNF4Aγ*$^{MI09417}$ experimental flies. *hsFLP*-induced cassette inversion in adult flies causes massive neurodegeneration, evident by the severe structural changes observed in brains of 3- and 10-day-old experimental animals (b and d), whereas age-matched control brains show no sign of neurodegeneration (a and c).

The following figure supplement is available for figure 4:

**Figure supplement 1.** P62 expression in *hsFLP*; *SNF4Aγ*$^{MI09417EGFP\text{-}T2A\text{-}mCherry}$/ *TM3,Sb[1]* animals.

# Materials and methods

## Plasmid construction

Flip-Flop constructs were generated using the following steps: 'attB-FRT14-FRT-SA-(GGS)4-PhaseX-EGFP-(GGS)4-SD-{SA-3xSTOP-SV40pA}reverse orientation-FRT14-FRT-attB' for all three reading frames were synthesised in pUC57 (Genewiz). The T2A-mCherry fragments with appropriate coding frame were amplified using PCR and cloned as AgeI/HindIII fragment in pUC57-"attB-FRT14-FRT-SA-(GGS)4-PhaseX-EGFP-(GGS)4-SD-{SA-3xSTOP-SV40pA} reverse orientation-FRT14-FRT-attB to create pFlip-Flop-P0, pFlip-Flop-P1 and pFlip-Flop-P2.

## Generation of Flip-Flop fly lines via RMCE

RMCE was performed as described in *Nagarkar-Jaiswal et al., 2015*.

## Mosaic analysis

Mosaics were generated in larval imaginal discs and brain tissue using *FLP* expressed under the control of eyeless promoter (*ey-FLP*) or using GAL4 drivers (*C155-GAL4* or *nSyb-GAL4*) to drive the expression of *UAS-FLP*. Corresponding crosses (*Figure 1—figure supplement 3*) were raised at 25°C. For experiments where *hsFLP* was used to generate mosaics, embryos from corresponding crosses (*Figure 1—figure supplement 3*) were collected for 24–30 hr. These embryos were given a heat-shock in a water bath at 37°C for one hr. Animals were raised at 25°C until the third larval instar stage before analysis. For adult mosaics, appropriate fly crosses (*Figure 1—figure supplement 2*) were set up at 18°C. Three-day-old adult flies were heat shocked at 37°C in an air incubator for 2 hr. For ovaries (*Figure 1—figure supplement 3c*), females were dissected 4 days after eclosion. For adult brains (*Figure 4B*), adults were recovered at 25°C and dissected on days 3 or 10.

## Immunostaining

Larval brain, imaginal discs, and adult brain staining were performed as described in *Nagarkar-Jaiswal et al., 2015*, adult ovary staining was performed as described in *Urban et al. (2014)*. Adult eye staining was performed as described in *Jaiswal et al. (2015)*.

## Antibodies

Primary antibodies: chicken anti-GFP 1:500 (Abcam, ab13970; RRID:AB_300798), rabbit anti-DsRed 1:500 (Clontech, 632496; RRID:AB_10013483), rabbit anti p62 1:2000 (*David-Morrison et al., 2016*) and rat anti-Elav 1:500 (DSHB, 7E8A10) (*O'Neill et al., 1994*). Secondary antibodies: Alexa 488 (RRID:AB_142924; Invitrogen, Life Technologies, Grand Island, NY), Cy5 (RRID:AB_2338072 and RRID:AB_2338393) and Cy3 (RRID:AB_2338059) conjugated secondary antibodies (Jackson ImmunoResearch, West Grove, PA) were used at 1:500. Phalloidin conjugated with Alexa 647 (Invitrogen) was used at 1:500.

## Electroretinogram (ERG)

Recordings were performed as described in *Nagarkar-Jaiswal et al. (2015)*.

## Adult brain histology

Adult fly heads were fixed in 8% glutaraldehyde (EM grade) and embedded in paraffin. Sections (10 μm) were prepared by a microtome (Leica) and stained with Hematoxylin and Eosin as described in *Chouhan et al., 2016*. At least three animals were examined for each genotype.

## Acknowledgements

We thank Yuchun He for microinjections. We thank Nele Haelterman, Oguz Kanca, David Li-Kroeger and Shinya Yamamoto for discussion and comments on the manuscript. This research was supported by NIGMS R01GM067858, whereas Confocal microscopy was supported by NICHD U54 HD083092 to the Baylor College of Medicine Intellectual and Developmental Disabilities Research Center. Stocks obtained from the Bloomington Drosophila Stock Center (NIH P40OD018537) were used in this study. HJB is an Investigator of the Howard Hughes Medical Institute.

## Additional information

### Competing interests

HJB: Reviewing editor, *eLife*. The other authors declare that no competing interests exist.

### Funding

| Funder | Grant reference number | Author |
| --- | --- | --- |
| Howard Hughes Medical Institute | | Sonal Nagarkar-Jaiswal Hugo J Bellen |
| National Institute of Neurological Disorders and Stroke | NINDS U54 NS093793 | Sathiya N Manivannan |
| Eunice Kennedy Shriver National Institute of Child Health and Human Development | NICHD U54 HD083092 | Zhongyuan Zuo |
| Robert A. and Renee E. Belfer Family Foundation | | Zhongyuan Zuo |
| National Institute of General Medical Sciences | R01GM067858 | Hugo J Bellen |

The funders had no role in study design, data collection and interpretation, or the decision to submit the work for publication.

### Author contributions

SN-J, Conceptualization, Data curation, Formal analysis, Validation, Investigation, Methodology, Writing—original draft, Writing—review and editing; SNM, Data curation, Formal analysis, Validation, Investigation, Methodology, Writing—original draft, Writing—review and editing; ZZ, Data curation, Methodology; HJB, Conceptualization, Formal analysis, Supervision, Funding acquisition, Writing—original draft, Project administration, Writing—review and editing

### Author ORCIDs

Sonal Nagarkar-Jaiswal, http://orcid.org/0000-0002-2369-3714

Sathiya N Manivannan, http://orcid.org/0000-0002-9470-2390

Hugo J Bellen, http://orcid.org/0000-0001-5992-5989

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
