## [Decision Letter]

Thank you for submitting your article "Flip-Flop, a facile, cell cycle-independent, conditional gene inactivation strategy differentially tagging wild-type and mutant cells" for consideration by *eLife*. Your article has been reviewed by two peer reviewers, and the evaluation has been overseen by a Reviewing Editor and K VijayRaghavan as the Senior Editor. The following individuals involved in review of your submission have agreed to reveal their identity: Michael Buszczak (Reviewer# 1), Frank Schnorrer (Reviewer #2).

The reviewers have discussed the reviews with one another and the Reviewing Editor has drafted this decision to help you prepare a revised submission.

Summary:

This paper describes a technique called Flip-Flop, which marks wild-type cells with EGFP-tagged endogenous protein and mutant cells with mCherry. Essentially the method relies on a newly designed MiMIC cassette. This cassette carries EGFP flanked by splice acceptor and donors and a T2A-mCherry flanked by a splice acceptor and multiple stop codons in the inverted orientation. Both fluorescent modules are flanked by FRT, FRT14 and AttB sites. Expressing flippase converts the protein trap into a mutagenic gene trap or vice versa. The authors use a number of specific examples to demonstrate the effectiveness of the technique.

The data are well-presented and the Flip-Flop technique will be useful to the research community, particularly in the clonal analysis of gene function in post-mitotic cells. This technique also enables the user to evaluate protein perdurance in mutant cells.

Essential revisions:

*Reviewer #1:*

One way to make this technique far more accessible to various researchers would be to generate a transgenic source of the three Flip-Flop cassettes. This would enable the targeting of currently available MiMIC lines with the Flip-Flop cassette using genetic crosses rather than injections. Is this possible?

The local chromatin environment may influence Flip-Flopping in different genes in different tissues. Do the authors see any evidence for this? The authors should provide quantification and statistics for the efficiency of Flip-Flopping in different settings, in both different genes and different tissues.

The authors should acknowledge/discuss that the level to which a GT can reduce gene function will depend on its position within a given gene and the efficiency of splicing. There may also be concerns that protein truncations will act in a dominant negative or neomorphic manner, potentially complicating the interpretation of results. Further discussion of the various advantages and disadvantages of the technique would strength the impact of the paper. Users of Flip-Flop will still have to characterize the degree to which a particular GT disrupts normal gene function.

*Reviewer #2:*

This new strategy uses the MiMIC collection and a new plasmid that can be inserted into the 2 attP sites of the MiMIC lines. It enables an irreversible Flp-induced conversion from a GFP protein trap, which should be functional, to an T2A-Cherry gene trap, which produces a mutant truncated protein, in addition to cytoplasmic Cherry from the endogenous enhancer. A nice by-product is the Cherry labelling of the mutant cells, at least if the gene studied is expressed at detectable levels.

The authors show that cassette inversion does indeed generate non-functional alleles for 4 genes, for which GFP fusions as viable. Thus, supposedly the tagged protein is functional and the gene trap is lethal. However, these studies were not done in clones.

For 2 of these genes (Trim 9 and SNF4A γ) they provide data for clonal loss of function phenotypes, in either all neurons during larval development for Trim9, resulting in sterility, or in all cells expressing SNF4a γ during development resulting in pupal lethality. For SNF4a they show that clonal loss of function during eye development also results in some defects. If loss of SNF4a γ is induced in all cells by heat shock specifically in adults the brain neurons degenerate.

Together, this suggests that the FlipFlop tool works well to induce loss of function phenotypes in a conditional manner for genes in which suitable MiMIC insertions exist. This allows conditional deletion of genes in the adult fly.

1) For this tool to be used by the broad community it would be important to know how many of the GFP-Cherry genetrap insertions result in functional proteins. As this cassette is much larger than the regular MiMIC exchange cassette I find this critical. The authors test only 6 genes, from which 4 result in viable flies and 2 do not. I would expect at least 10 or 20 cassette exchanges to provide statistics on this important point. By the cassette exchange method using transgenes, as published by the authors in 2016 in *eLife*, it should be easy to do it for a larger number of genes.

2) I am a bit disappointed by the phenotypical analysis done in this study. The authors don't use the power of the system, having mutant and wild-type cells next to each other to assess cellular phenotypes. The reason for that might be lack of time – see point 3.

3) The Clandinin lab used a very similar Flp induced method (called FlpStop) to also generate conditional alleles with MiMIC lines. The Flip inversion method appears very similar. Differences are that the wild-type protein is not labelled in the wild type cells and the mutant cells are labelled by UAS-tomato rather than gene trap Cherry (the former might be of advantage at least if the protein of interest is expressed a low level). This other study induced conditional alleles for 6 genes showing that the strategy indeed appears to work generally. As this study was published in February 2017 in *eLife* (Fisher et al.) the authors should compare the 2 strategies at least in the Discussion of a revised version.

---

## [Author Response]

*Essential revisions:*

Reviewer #1:

*One way to make this technique far more accessible to various researchers would be to generate a transgenic source of the three Flip-Flop cassettes. This would enable the targeting of currently available MiMIC lines with the Flip-Flop cassette using genetic crosses rather than injections. Is this possible?*

Yes, this is possible. We recognize that using a genetically deliverable donor for the Flip-Flop cassette will be useful to the community. We have earlier developed such a tool for EGFP-FlAsH-StrepII-TEVcs-3xFlag {GFSTF} tag used for RMCE with existing MiMICs (Nagarkar-Jaiswal et al., 2015 *eLife*; 4:e08469). This system involved the use of direct FRT repeats to circularize an RMCE cassette in vivo and use it as a donor in a PhiC31 mediated cassette exchange. However, the Flip-Flop cassette uses FRT sites for cassette inversion. Hence, we will need a Cre/LoxP system to generate the donor as done by Diao et al., (Diao et al., 2015). However, cloning of the LoxP sites in a construct with several repetitive elements, creating at least 6-10 transgenes, testing many MiMICS and determining the efficiency will take at least 6-8 months. We will create the genetic donors in the future but it is beyond the scope of this proof-of-principle study.

*The local chromatin environment may influence Flip-Flopping in different genes in different tissues. Do the authors see any evidence for this? The authors should provide quantification and statistics for the efficiency of Flip-Flopping in different settings, in both different genes and different tissues.*

We agree with the reviewer that cassette inversion will depend on the accessibility of FRT sites for the Flippase. To examine this in a quantitative fashion for multiple genes, we have extended our initial analysis of Flip-Flop inversion for six genes using the same source of Flippase in identical culture conditions. This analysis reveals that inversion in eye-antennal discs is as follows: for *SNF4Aγ, Trim9* and *eff* the efficiency is more than 95%, whereas for *Nedd8, cdep* and *Sik3* it is higher than 70%. We present these data in Figure 1—figure supplement 5. We have also modified the text according to the reviewer’s suggestion.

*The authors should acknowledge/discuss that the level to which a GT can reduce gene function will depend on its position within a given gene and the efficiency of splicing. There may also be concerns that protein truncations will act in a dominant negative or neomorphic manner, potentially complicating the interpretation of results. Further discussion of the various advantages and disadvantages of the technique would strength the impact of the paper. Users of Flip-Flop will still have to characterize the degree to which a particular GT disrupts normal gene function.*

We thank the reviewer for raising this concern. We would like to point out that the splice acceptor site used in Flip-Flop cassette is the same as in the original MiMIC cassette. In our previous study (Nagarkar-Jaiswal et al., 2015, *eLife*), we found that 92% of the MiMIC gene trap insertions are severe loss of function alleles of the gene of interest. In contrast, none of the MiMIC insertions in the reverse orientation (non-GT) caused an obvious phenotype. In addition, we checked for aberrant splicing and found no exon skipping in three out of three genes. In our current study, we evaluated the effect of the GT orientations by generating GT-oriented Flip-Flop lines and examined their ability to complement a deficiency. Seven out of nine complemented a deficiency and the encoded protein is tagged with GFP, whereas two GFP insertions failed to complement the def. This is in line with our previously published data where 77% of the tagged proteins with GFP were functional (Nagakar-Jaiswal et al., 2015). Upon Flip-Flop the viable GFP tagged genes became tagged with RFP and all became lethal when tested as homozygotes or over a deficiency, indicating that they are loss of function mutations. Importantly, none of the heterozygous lines show a dominant phenotype, suggesting that we do not induce dominant negative or dominant gain of function mutations. We also did not observe any dominant phenotypes in the 58 strains tested in Nagarkar-Jaiswal et al. that carried SA-polyA tails that caused truncations. In summary, the vast majority of the GT-lines are strong loss of function alleles and do not show a neomorphic/gain-of-function or dominant negative phenotype that could arise from a truncated, yet lingering protein with a dominant negative molecular function. We are aware of the issues raised by the reviewer and will mention some of the above data in the Discussion.

Reviewer #2:

*[…] 1) For this tool to be used by the broad community it would be important to know how many of the GFP-Cherry genetrap insertions result in functional proteins. As this cassette is much larger than the regular MiMIC exchange cassette I find this critical. The authors test only 6 genes, from which 4 result in viable flies and 2 do not. I would expect at least 10 or 20 cassette exchanges to provide statistics on this important point. By the cassette exchange method using transgenes, as published by the authors in 2016 in eLife, it should be easy to do it for a larger number of genes.*

The MiMIC cassette is 1400 nt and the current one is 2600 nt. It is not a very big difference and the size does not seem to be an issue. Indeed, we compared the protein traps generated using the previous construct (Venken et al., 2011) and the Flip-Flop cassette for all the lines described in this manuscript. We didn’t find any obvious difference between the expression pattern and the genetics based on complementation tests. If the protein traps are functional with the GFP cassette from Venken et al., they are also functional with GFP derived from the Flip-Flop, and complement each other. This argues that there are no obvious differences between the two tagging methods and we now discuss this finding.

To accommodate the reviewer’s concern, we have included three more genes that we were in the process of generating prior to submission (*Ctrip, Eip63E* and *Ank2*). We have no other conversions and it would take us many more months to create and test more lines.

*2) I am a bit disappointed by the phenotypical analysis done in this study. The authors don't use the power of the system, having mutant and wild-type cells next to each other to assess cellular phenotypes. The reason for that might be lack of time – see point 3.*

To satisfy the reviewer we incorporated new data on SNF4Ay. The data are new. They are included in the Results section and we have included them in Figure 4.

*3) The Clandinin lab used a very similar Flp induced method (called FlpStop) to also generate conditional alleles with MiMIC lines. The Flip inversion method appears very similar. Differences are that the wild-type protein is not labelled in the wild type cells and the mutant cells are labelled by UAS-tomato rather than gene trap Cherry (the former might be of advantage at least if the protein of interest is expressed a low level). This other study induced conditional alleles for 6 genes showing that the strategy indeed appears to work generally. As this study was published in February 2017 in eLife (Fisher et al.) the authors should compare the 2 strategies at least in the Discussion of a revised version.*

We agree that the two strategies employ similar Flex switch to induce local cassette inversion. However, there are main differences between the two studies. Fisher et al., do not tag the endogenous protein with a tag. While this may seem beneficial for genes where an internal tag is deleterious to the protein, we argue that for a clear majority of the genes the internal EGFP tag is not detrimental. The major advantage to the internal tag is the ability to track the expression of the gene of interest. Moreover, it allows anti-GFP antibody mediated immunoprecipitations followed by mass spectroscopy (David-Morrison et al., 2016; Yoon et al., 2017), enables ChIP sequencing (Negre et al., 2011), and permits mRNA or protein removal with iGFPi (Neumuller et al., 2012) or deGradFP (Caussinus et al., 2011; Nagarkar-Jaiswal et al., 2015; Urban et al., 2014). Finally, the EGFP tag allows us for the first time to track protein perdurance in mutant cells.

The second major difference is the expression of mCherry to mark the mutant cells. In Fisher et al. tdTomato is expressed under the control of the UAS/GAL4 system. Therefore, cells that have undergone cassette flip express tdTomato irrespective of the gene of interest’s expression. This could dilute the effect of true-mutant cells with cells where cassette inversion has occurred but do not bear any consequence to the gene of interest. In contrast, the Flip-Flop technology allows cells to be marked with mCherry only where the cells would otherwise express the gene of interest (Figure 5).

Author response image 1.Contrast between the FLPstop and Flip-Flop technologies.(Top) A hypothetical gene expression in a wing imaginal disc recapitulated by internally tagged EGFP expression from the Flip-Flop cassette (green). Upon flippase expression mCherry expressing mutant cells are generated within the domain of the gene’s expression. (Botton) FlpStop construct does not provide any information on the expression of the gene. Mosaics generated could lie outside the domain of the gene’s expression and therefore may not be relevant.**DOI:**
http://dx.doi.org/10.7554/eLife.26420.016

We recognize that the mCherry expression under UAS/GAL4 may be beneficial as it provides a bright read-out compared to that of the T2A mCherry which depends on the gene’s expressivity to be detected. We argue that the loss of EGFP and/or the mCherry together permit the evaluation of the mutant cells and thus the level of mCherry expression alone is not a limiting factor for detecting mutants.